# Influence of Nutritional Status and Physical Fitness on Cognitive Domains among Older Adults: A Cross-Sectional Study

**DOI:** 10.3390/healthcare11222963

**Published:** 2023-11-15

**Authors:** Carmen Boquete-Pumar, Francisco Álvarez-Salvago, Antonio Martínez-Amat, Cristina Molina-García, Manuel De Diego-Moreno, José Daniel Jiménez-García

**Affiliations:** 1Department of Health Sciences, Faculty of Health Sciences, University of Jaén, 23071 Jaén, Spain; carmenbp.ef@gmail.com (C.B.-P.); amamat@ujaen.es (A.M.-A.); manueldediego@eade.es (M.D.D.-M.); josedanieljimenezgarcia@gmail.com (J.D.J.-G.); 2Department of Physiotherapy, Faculty of Health Sciences, European University of Valencia, 46010 Valencia, Spain; 3Health Sciences PhD Program, Universidad Católica de Murcia UCAM, Campus de los Jerónimos nº135, 30107 Guadalupe, Spain; cmolina799@ucam.edu

**Keywords:** nutrition status, exercise test, cognitive function, aged individuals

## Abstract

Background: The health issues presented by the aging population can result in reduced muscle mass, poorer physical function, and cognitive impairment. The goal of this study was to determine how nutritional status and physical fitness relate to cognitive impairment in older adults. Methods: A cross-sectional descriptive and analytical study involving 100 participants was carried out to analyze the impact of nutritional status and physical fitness on cognitive impairment. Nutritional status was assessed with the Eating Assessment Tool (EAT-10) and The Mini Nutritional Assessment—Short Form (MNA-SF); physical fitness via the implementation of manual grip evaluation, the 4-m walking test (4-MWT), and the Timed Up and Go (TUG) test; and cognitive impairment evaluation was conducted using the Mini-Mental State Examination (MMSE), the Boston Naming Test (BNT) and the Controlled Oral Word Association Test (COWAT). Results: Data analysis revealed that higher malnutrition status was associated with fewer correct responses in the COWAT (R^2^ = 0.421), while a correlation between higher MMSE and BNT scores, faster completion times for the 4-m walking and TUG tests, and an increase in handgrip strength load was also observed. Conclusion: The analysis of the data revealed that those individuals with superior nutritional status and greater levels of physical fitness outperformed others on the cognitive evaluation.

## 1. Introduction

Along with many social and personal challenges, both population aging and musculoskeletal diseases have become serious public health problems [1]. Due to the increase in life expectancy, the need for studies that shed light on the health problems presented by the aging population arises. Aging goes hand in hand with certain changes, which can lead to decreased muscle mass, decreased physical function, and cognitive impairment [2]. This last issue is a growing public health concern and is thought to be directly related to environmental risk factors, including diet and physical activity [3,4]. It is estimated that 0.75–3% of adults over 65 years of age will develop cognitive impairments that may reduce their quality of life by reducing their work and social skills [5]. Mild cognitive impairment (MCI) is considered the intermediate stage between the variations observed in normal cognitive aging and the biological changes associated with dementia [4]; older adults with MCI have a higher risk of progression to dementia, although more than 50% remain stable or even return to normal [6]. If cognitive impairment and/or dementia could be delayed by 5 years, its prevalence would be reduced by half [3]; therefore, in the absence of a curative treatment, preventing or postponing the onset of this condition is of vital importance.

According to several studies, nutritional factors [3,5,7] and the physical condition of participants [1,4,8] were shown to be related to lower manifestations of dementia and cognitive impairment. Although these studies’ findings are very encouraging, it is still unclear how many aspects of lifestyle interact with one another as cognitive impairment progresses [9]. When discussing diet and aging, it is important to emphasize that presenting a deficient nutritional status, nowadays, constitutes one of the most frequent problems in the elderly population and tends to be overlooked [10]. The nutritional status of an individual is understood as the result of the balance between nutritional intake and nutritional requirements [11] and the state of malnutrition is characterized by a multifactorial nature, associated with physical, physiological, psychological, and social aspects [3]. Among these factors, the relationship between dysphagia and undernutrition, as well as dietary consumption patterns in older adults [12], appear to be extremely relevant.

Exploring physical fitness parameters among older adults, it must also be taken into consideration that sarcopenia is a loss of skeletal muscle mass and physical function (muscle strength or physical performance) that occurs with advancing age, associated with significant personal and economic costs, and is increasingly common in this population [13,14]. Preventing age-related loss of muscle mass and strength is critical to preserving physical capacity and achieving independent living in older adults. The European Working Group on Sarcopenia in Older People (EWGSOP) acknowledges that strength is a remarkable indicator of future autonomy loss compared to the measurement of skeletal muscle mass. The EWGSOP advises employing a variety of methods for evaluating older individuals, including the handgrip strength test, gait speed test, and the Timed Up and Go (TUG) test, among others [15].

Since preventive lifestyle practices are widely employed, little is known about the interactions between nutrition markers, physical performance, and cognitive impairment in older adults [9]. Considering these findings, researching the possible connections between nutritional values (analyzing the nutritional status and level of dysphagia), physical condition (evaluating upper body strength, gait speed, and dynamic balance), and a lower deterioration of cognitive level may present a very useful preventive tool in this population.

This study was designed to investigate how older adults’ nutritional status and physical fitness (upper extremity strength, gait speed, and dynamic balance) relate to cognitive impairment in the population aged 60 years old and over.

## 2. Materials and Methods

### 2.1. Study Design and Participants

A cross-sectional descriptive and analytical investigation was conducted, involving a total of 100 older adults aged over 60 years old who were randomly chosen through the departments of Social Affairs of the City Councils of Pizarra (Málaga) and Herrera (Seville) (Figure 1). To be eligible for inclusion, participants had to be over 60 years old, listed as residents on the municipalities of Pizarra and Herrera, and reading, writing, and speaking Spanish. Having central or peripheral neurological alterations, severe cognitive impairment, serious somatic or psychiatric diseases, rheumatic diseases, pacemakers or prostheses, or any type of alteration or pathology that could affect balance and functional activity (such as auditory or vestibular alterations) were all considered to be exclusion criteria. The University of Jaen’s Ethics Committee granted approval for this study on 19 February 2018, under protocol code DIC.17/5.TES, and it was carried out in accordance with the Declaration of Helsinki’s principles. Prior to the study’s implementation, each participant signed an informed consent form. Each evaluation was conducted by a previously trained evaluator and included a personal interview as well as the completion of the assessment tests.

The study sample size was estimated using the formula for estimating proportion, where the predicted rate was 0.3, with *p* = 0.05 and a precision of ±0.06 in the study population. By including 25% to account for human error, a necessary sample size of 179 patients was acquired to account for transcription errors, exclusions, and unwilling participants. To maintain the reference sample size (this was determined using the G*Power software, Version 3.1.9.7, Axel Buchner, Heinrich-Heine-Universität Düsseldorf, Düsseldorf, Germany), when a chosen person could not be included in the sample due to an absence of access to it, lack of predisposition, or failure to meet the inclusion criteria, they were replaced by the next person on the list. After the sample was collected and the recruitment process was over, participants and their families were given presentations to explain the program and its operation.

### 2.2. Study Outcomes

#### 2.2.1. Sociodemographic and Anthropometric Data

Records of sociodemographic information included the person’s full name, age, birthdate, marital status, level of education, and location of residence. Anthropometric data (body mass index [BMI], waist–hip ratio [WHR], height, and weight), as well as habits such as the quantity and nature of physical activity, smoking, and recent falls, were also gathered.

Additionally, two measurements of the abdominal perimeter (waist) were made of each subject in a standing position, using a 1.5 m flexible tape (Lufkin, W606PM, Harford County, MD, USA) and taking the equidistant point between the last rib and the hip (iliac crest) as the reference.

#### 2.2.2. Nutritional Status

Dysphagia and malnutrition were the markers used to gauge the participants’ nutritional status. The Eating Assessment Tool (EAT-10), which has been validated in Spain [16], is composed of 10 items, specifically to assess the risk of dysphagia. In each of the questions, the individual’s ability to swallow in different situations is evaluated from zero to four, with zero being no problem and four being a severe problem [16].

The Mini Nutritional Assessment—Short Form (MNA-SF) questionnaire is a short and valid nutritional screening tool for elderly populations [17]. The MNA-SF is a six-item questionnaire that considers food intake, weight loss, and physical or psychological stress during the last 3 months, as well as body mass index [18], and has been validated in Spain [19]. Based on the score obtained in the MNA-SF, there are three possible classifications: from 0 to 7 points (malnutrition), from 8 to 11 points (risk of malnutrition), or from 12 to 14 (normal nutritional status) [17].

#### 2.2.3. Physical Fitness

The dimensions that were evaluated for the evaluation of the participant’s physical performance were upper extremity muscle strength, gait speed, and dynamic balance.

A manual grip evaluation was performed by making use of a manual hydraulic dynamometer (TKK 5401 Grip-D, Takey, Tokyo, Japan), usually applied in these kinds of investigations to measure upper extremity muscle strength. Participants performed three maximal grip strength attempts with their dominant hand, and a 30 s break was taken in between each one. To ensure an appropriate grip, the dynamometer was set to 5.5 sizes for males, and for females the dynamometer was adjusted with respect to the size of the hand [20].

The 4-m walking test (4-MWT) was used to measure gait speed. It has been claimed that gait speed, assessed at a person’s regular pace, is a useful clinical marker of health, well-being, and functional status in the elderly population [21]. The starting and ending positions were marked with a 4 m line on the floor. Participants were encouraged to maintain their normal and comfortable speed while walking until they reached the finish line [22]; the first attempt was carried out to ensure familiarity with the test, while the time taken to perform the second trial was tracked and collected by the evaluator. The participant’s foot contacting the ground at the beginning and conclusion of the walking course signaled the beginning and end of timekeeping, respectively. Walking speed was estimated by dividing the distance by the time required to cover the distance (m/s). The 4-MWT demonstrated exceptional concurrent validity and test–retest reliability: intraclass correlation coefficient (ICC) = 0.991 (95% CI: 0.983, 0.996); standard error of measurement (SEM) = 0.032; and minimal detectable change-95% (MDC-95%) = 0.090 m/s [23].

The functional evaluation used to assess dynamic balance was the Timed Up and Go (TUG) test. The procedure entails measuring the time taken by the participant to get out of a chair, walk a 2.44 m distance, turn around, and get back in. Each participant took three tries, with the fastest attempt being the one retained by the evaluator. The TUG test has been shown to be an accurate and reliable method of determining geriatric mobility and fall risk, and it has been widely applied in both clinical and research contexts. It is specifically considered as a reliable and accurate tool for evaluating dynamic balance and risk of falling in aging individuals around the world [24].

#### 2.2.4. Cognitive Impairment

The tools used to determine whether the person had any cognitive impairment consisted of three tests.

The first one, the Mini-Mental State Examination (MMSE), is a cognitive test that is frequently employed in the assessment of possible dementia [25]. The MMSE is divided into two sections, the first of which has a maximum score of 21 and solely calls for verbal responses to questions like orientation, memory, and attention. The second section evaluates the ability to name, obey verbal and written instructions, write a whole phrase on the spot, and accurately copy a complex polygon resembling a Bender-Gestalt Figure. The top score is nine. Patients with significant vision impairment may experience some additional difficulty in Part II because of the reading and writing requirements, although this difficulty is typically mitigated by large writing and considered when scoring. The maximum possible score is 30, and normal cognition is indicated by any score of 24 or higher. Scores below this point may suggest cognitive impairment that could be severe (≤9 points), moderate (10–18 points), or mild (19–23 points).

The second test implemented was the Boston Naming Test (BNT). This test is currently a prevalent visual confrontation naming test for aphasia, dementia, and other geriatric conditions. The current version of this test is a 60-item model [26], which was trimmed from the original 85 items and is more commonly utilized. There is also the Shortened Boston Version-15 (SBV-15) proposed by Mack and colleagues [27], which is correlated with the full 60-item form and might thus take the place of the latter in clinical settings when time constraints, unique participant characteristics, or both call for the use of a shorter version [28] as in the case of the current study. In the BNT, participants are asked to verbally name fifteen line drawings of everyday objects: five of the items are simple, five are moderately challenging, and five are challenging. Within 20 s of administration, a spontaneous response must be given; if not, two prompting signals (one phonemic and one semantic) may be utilized. The scoring process considers the number of accurate responses that were generated spontaneously, the number of cues that were provided, and the number of responses after phonemic and semantic cuing.

The third test employed was the Controlled Oral Word Association Test (COWAT), used to measure phonologic and semantic fluency [29]. The COWAT has two components: a phonologic fluency test and a semantic fluency test. The phonologic assessment evaluates the ability to produce words that fall into the same category or that start with a specific letter on demand. In the semantic fluency evaluation, participants receive an instruction to quickly come up with the names of as many animals as they can. For 60 s, the subject must speak as many words as they can that start with a specific letter of the alphabet; the score of the test is the total amount of words generated, and no proper names or word elaborations are allowed [30].

### 2.3. Statistical Analysis

Means and standard deviations were used to summarize continuous variables, whereas percentages as well as frequencies were chosen to summarize categorical variables. To determine whether all variables had a normal distribution, the Kolmogorov–Smirnov test was used.

The potential individual relationships between independent variables like nutritional status, lower limb strength, upper extremity muscle strength, and gait parameters, as well as other covariates consisting of BMI, sex, age, and educational attainment level, were examined utilizing a bivariate correlation analysis to ascertain their relationship with cognitive impairment. The Pearson coefficient was used. A step-by-step approach and a multivariate linear regression model were both utilized to add variables to the model as a means of assessing any potential independent relationships between the research variables. Cognitive impairment was included in the multivariate linear regression as a dependent variable in various models (significant in bivariate correlation, “*p* < 0.05”). The effect size coefficient of multiple determination in linear models was computed using adjusted R2. R2 was categorized as negligible if <0.02, medium if 0.02 and 0.15, and big if >0.35. The 95% confidence interval was utilized (*p* < 0.05). Data analysis was conducted using the SPSS statistical software (version 27.0) for Windows (SPSS Inc., Chicago, IL, USA).

## 3. Results

In a broader sense, 100 senior citizens (mean 67.05 ± 5.17 years, between 60 and 75 years old) participated in the present study. Most of the participants had primary education or below (79.00%) and the mean BMI was 29.70 ± 4.91 kg/m^2^. The descriptive data of the variables analyzed in this study, displayed in Table 1, showed that the total scores in nutritional status were 0.64 ± 1.48 for dysphagia and 12.28 ± 1.80 for malnutrition. With respect to gait speed, the time taken to walk 4 m was 2.27 ± 0.48 s. Pertaining to strength levels, handgrip strength was 24.60 ± 7.38 kg and time to realize TUG test was 7.07 ± 1.53 s. Concerning the dependent variables, the scores for the screener for dementia were 25.96 ± 3.20 for the MMSE and 10.27 ± 2.93 for language. The descriptive data of the other dependent variables analyzed in this study showed that the number of words were 28.35 for semantic fluency ± 10.83 w and 32.07 ± 12.08 w for phonologic fluency.

The bivariate analysis (Table 2) showed that the dependent variables of the present work screener for dementia, language, and semantic and phonologic fluency showed significant positive correlations with educational status. An age-related substantial negative relationship with the overall score on the screener for dementia and language and the total words in phonological fluency was also found. Furthermore, participants’ sex was related to language and phonologic fluency. This last variable was related to nutritional health, most specifically the MNA-SF and dynamometer-measured strength levels. Regarding the total words measured with semantic fluency, significant negative correlations were also observed with gait speed and dynamic balance. An existing correlation was found between a higher MMSE score and a shorter time required to complete the 4 m walking and TUG tests. A higher BNT score was correlated with a shorter time to complete the 4 m walk and TUG tests, as well as an increase in the load (kilograms) of handgrip strength.

The results of the multivariate linear regression analysis displayed in Table 3 reveals several distinct connections between various cognitive status variables analyzed in this study. Higher educational levels were associated with better MMSE and BNT scores, as well as a more elevated total recount of words in P-COWAT and S-COWAT (R^2^ = 0.165; 0.380; 0.421; and 0.196). Relating to age, younger participants were associated with better BNT scores and P-COWAT answers with more total words counted (R^2^ = 0.380 and 0.421). Referring to nutritional status, a higher malnutrition status had an association with decreased word responses in the P-COWAT (R^2^ = 0.421). Lower gait speed times on the 4 m walking test were connected to higher overall MMSE and BNT scores (R^2^ = 0.165 and 0.380), and with a higher total word response in the P-COWAT (R^2^ = 0.196). A shorter time to complete TUG test was related to a higher score in the MMSE (R^2^ = 0.165).

## 4. Discussion

This study aimed to explore possible associations between older individuals’ nutritional status, physical fitness (upper extremity strength, gait speed, and dynamic balance), and cognitive impairment. The study’s major findings, which covered 100 individuals who were aged over 60, suggest substantial correlations between the different independent variables and cognitive status.

The present investigation’s bivariate analysis revealed a positive association between educational status and the dependent variables of cognitive impairment, including language, semantic fluency, and phonologic fluency. There was also evidence of an age-related negative connection between the overall score on the screener for dementia and language and the total words shown in phonological fluency. These results are consistent with previous investigations that demonstrated a common correlation, at least to some degree, between older people’s performance on neuropsychological tests and demographic variables such as advanced age or a lack of formal education [31,32]. In addition, semantic and phonologic fluency were associated with the participant’s sex; this is in line with previous results showing that women may slightly outperform males on this verbal fluency task [33]. On the other hand, yet another study found no appreciable gender difference in the semantic fluency results, only that male generated more animals’ names than females without this being of significant value [34].

The current study’s phonologic fluency was related to nutritional status, most specifically MNA-SF and dynamometer-measured strength levels. A previous study [35] has indicated that the participants’ nutritional status (assessed by the MNA-SF and blood tests) appears to be an indicator of their cognitive status (screened by MMSE), proving that those with a normal nutritional status according to the MNA-SF presented with an intact cognitive status. In general, as determined by a 2020 Pooled Analysis Study [36], there was a substantial correlation between improved physical function performance, including walking speed, chair rise, and dynamic balance, and higher scores on the MMSE. When analyzing the studies included in this research, the MMSE had no significant connection with hand grip strength. The only study on this analysis that presented a strong association between the MMSE and physical performance was a double-blind, randomized, placebo-controlled investigation in which the experimental group was taking nutritional supplementation [37], which could suggest a better nutritional status. However, some studies have shown that older adults with a weaker grip in their hands were more likely to have cognitive impairments than those with a stronger grip [38,39]. Looking into the total of words measured with semantic fluency, significant negative correlations were also observed with gait speed and dynamic balance.

An existing correlation was found between higher MMSE and BNT scores and a shorter time required to complete the 4 m walking and TUG tests, as well as an increase in the load (kilograms) of handgrip strength. In relation to walking speed, prior research found that in a sample of older persons, those with mild cognitive impairment and the ones with Alzheimer’s disease exhibited a slower speed compared with cognitively healthy ones [40]. In this same study, the TUG test times were worse for those participants with Alzheimer’s disease when compared to the ones with mild cognitive impairment or who were cognitively healthy. It is significant to point out that the patients in this study had prior diagnoses of their cognitive impairment level, and no evidence related to upper extremity strength or nutritional status was included. To the best of our knowledge, and except for one study already named [37], there has not been much research combining nutritional status with upper body strength, gait speed, dynamic balance, and their impact on cognitive impairment in the same sample. Contemplating this, and especially when it comes to the clinical implications of our results, our paper addresses how each component examined may alter older adults’ cognitive impairment delay and, consequently, their quality of life. Future studies should therefore put focus on using a combined system of nutritional and physical parameters to address cognitive impairment.

Along with the above-mentioned aspects, the multivariate linear regression analysis revealed that higher educational levels were associated with better MMSE and BNT scores, as well as an elevated total recount of words in the P-COWAT and S-COWAT (R^2^ = 0.165; 0.380; 0.421; and 0.196). These results are in line with the ones presented in a previous study that aimed to determine whether people with amnestic mild cognitive impairment show more rapid cognitive loss than those with lower levels of education [41], which concluded that higher educational attainment is associated with a reduced incidence of Alzheimer’s disease than lower educational attainment. With relation to age, younger participants obtained superior BNT scores and P-COWAT replies with more words overall when compared to older participants. These results are supported by earlier evidence which shows that a significantly lower mean on the BNT scores and greater variability are associated with increasing age [42]. Additionally, this meta-analysis demonstrated that aging had an impact on COWAT performance for phonemic verbal fluency, suggesting a decline [43]. Furthermore, this study concludes that more investigation based on cross-sectional designs and similar age comparisons should be undertaken, a fact that we also reiterate.

Regarding nutritional status, a higher malnutrition status had a connection with reduced word responses in the P-COWAT (R^2^ = 0.421); it should be mentioned that no study that measures and relates the same variables in this manner was found in the search for material to support these results, but they could be connected somehow to a cross-sectional study that deals with nutritional status and its association with behavioral psychological symptoms of dementia (BPSDs) [44]. It is crucial to note that nutritional status was evaluated using the MNA-SF and global cognitive function was examined by using the MMSE in the same manner as we did, but they also assessed the BPSDs by using the Dementia Behavior Disturbance Scale (DBD). While well-nourished subjects and those at risk of or experiencing malnutrition did not differ in their MMSE scores, this likewise occurred during our statistical analysis, but the latter group displayed significantly higher DBD scores (particularly in terms of verbal aggression, emotional disinhibition, apathy, and memory impairment) and lower MNA-SF scores. These findings also point to the prevalence of malnutrition among older persons with mild cognitive decline, but also suggest that nutritional problems may be linked to certain BPSDs. As opposed to our results, another cross-sectional study conducted in July 2021 [45] found no relationship between cognitive frailty and nutritional status (malnutrition), but it is of the utmost importance to indicate that the evaluation methods for both parameters were assessed differently compared to in our study. Prealbumin serum was used to measure nutritional status and the Montreal Cognitive Assessment-Basic (MoCA-B) was used for cognitive function. Therefore, future investigations should continue to consider nutrition as another fundamental key component in the approach to these patients and should additionally attempt to homogenize the evaluation tools to make it simpler to generalize the data acquired.

According to the last outcome of the multivariate linear regression analysis, a shorter TUG test completion time was related to a higher MMSE score (R^2^ = 0.165). Considering that lower times on the TUG test were correlated with a more favorable dynamic balance and higher scores on the MMSE reflect normal cognition, it could be asserted that those participants with an optimal dynamic balance presented a lower level of cognitive impairment. A cross-sectional observational study that used the same two tests to detect dynamic balance and cognitive impairment, respectively, was shown to be in line with the results reported above [46]. This same study emphasizes the value of further research into the factors underlying these relationships, such as the significance of the brain mechanisms. For our part, the direction to take is the same because it would be interesting to see how the brain mechanisms connected to a motor task can affect the various brain regions involved, if they can be stimulated, how this might impact physical performance, and how it all in conjunction might affect cognitive function. Also, the findings presented in this study could help the scientific community create intervention programs that focus on enhancing physical and nutritional health to prevent cognitive impairment at the clinical level.

### Strengths and Limitations

As far as we are aware, this study is the first to compare nutritional status, physical fitness, and cognitive impairment in the same group of older adults. It is essential to bear in mind that the tools of evaluation adopted for each variable were precisely selected based on the characteristics of the population and are widely used instruments for assessing the suggested objective [16,17,18,19,20,21,22,23,24,25,26,27,28,29,30]. However, there are certain limitations that are important to acknowledge. Any generalization of the results should be restricted to people with similar characteristics because the cross-sectional design of the study means that casual relationships cannot be ruled out, and because it was conducted among men and women who were 60 years of age or older from two specific geographic areas. The results should be interpreted with caution because no prior level of physical activity was considered for participation, and it is possible that those who engage in greater physical activity perform better on cognitive tests [46].

Although at the beginning it seemed interesting to establish a relationship between dysphagia and cognitive impairment, this association did not show significant results. Given that the MNA-SF is a screening tool, the results should be interpreted with caution, and future investigations should consider using a more accurate assessment tool, such as the Mini Nutritional Assessment—Long Form (MNA-LF). For there to be sufficient evidence to further confirm the casual association found, deeper research will be required to enlarge the sample size, follow up with this cohort, add more information about food intake frequency, macronutrients, and blood test results, and validate these findings in other populations.

## 5. Conclusions

Our study analyzed the serial multiple mediation roles of nutritional status and physical fitness on the relationship between cognitive impairment in older adults aged 60 years old or above. A breakdown of the results shows that participants who had an optimal nutritional status (assessed by the MNA-SF) and higher scores of physical fitness (hand grip, gait speed, and dynamic balance) performed better on the cognitive evaluation (MMSE, BNT, and COWAT). Therefore, these nutritional, physical, and cognitive variables should be taken into consideration to manage and prevent cognitive impairment in this population, given that they may significantly impact their quality of life. Additionally, at the clinical level, the research community may be able to develop intervention programs with the aid of the findings reported in this study.

## Figures and Tables

**Figure 1 healthcare-11-02963-f001:**
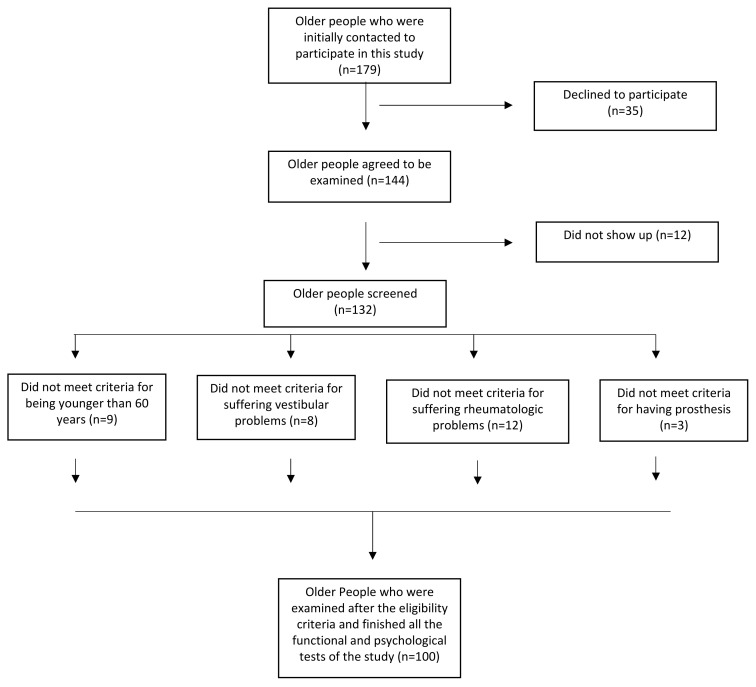
Flow diagram of study participants.

**Table 1 healthcare-11-02963-t001:** Descriptive data of the sample (n = 100).

Characteristics		Values Total = 100		Values	Men = 21	Values	Women = 79
		FREQUENCY	%	FREQUENCY	%	FREQUENCY	%
Educational status, n (%)	No formal education	14	14	2	9.52	12	15.18
	Primary education	65	65	16	76.19	49	60.02
	Secondary education	17	17	2	9.52	15	19.98
	University	4	4	1	4.76	3	3.79
		MEAN	SD	MEAN	SD	MEAN	SD
Age (years)		67.5	5.67	64.80	8.59	69.56	7.89
BMI (kg/m^2^)		29.70	4.91	29.63	4.18	29.46	5.60
EAT-10 (score)		0.64	1.48	0.50	0.79	0.65	1.56
MNA-SF (score)		12.28	1.80	12.25	1.95	12.28	1.80
Handgrip (kg)		24.60	7.38	37.18	7.66	22.25	4.32
Gait speed (s)		2.27	0.48	2.06	0.45	2.31	0.48
Dynamic balance (s)		7.07	1.53	6.66	1.46	7.15	1.54
MMSE (score)		25.96	3.20	26.66	2.47	25.83	5.05
BNT (score)		10.27	2.93	12.00	2.51	9.95	2.91
P-COWAT (words)		30.24	12.78	36.52	12.18	29.07	5.05
S-COWAT (words)		31.93	9.85	34.76	13.30	31.43	9.04

BMI: Body Mass Index. EAT-10: Eating Assessment Tool. MNA-SF: Mini Nutritional Assessment—Short Form. MMSE: Mini Mental State Examination. BNT: Boston Naming Test. S-COWAT: Semantic fluency. P-COWAT: Phonologic fluency.

**Table 2 healthcare-11-02963-t002:** Pearson’s correlations between variables analyzed in this study.

	MMSE (s)	BNT (Score)	P-COWA (Words)	S-COWA (Words)
EAT-10 (score)	0.019	−0.037	0.049	−0.089
MNA-SF(score)	−0.099	0.004	−0.219 ^1^	0.064
Handgrip (kg)	0.156	0.273 ^2^	0.271 ^1^	0.142
Gait Speed (s)	−0.206 ^1^	−0.331 ^2^	−0.121	−0.358 ^2^
Dynamic Balance(s)	−0.272 ^2^	−0.240 ^2^	−0.139	−0.304 ^2^
Sex	−0.096	−0.255 ^2^	−0.213 ^1^	−0.124
Educational Status	0.341 ^2^	0.523 ^2^	0.555 ^2^	−0.324 ^2^
Age (years)	−0.214 ^1^	−0.404 ^2^	−0.411 ^2^	−0.154
BMI (kg/m^2^)	−0.116	−0.089	−0.063	−0.115

BMI: Body Mass Index. EAT-10: Eating Assessment Tool. MNA-SF: Mini Nutritional Assessment—Short Form. MMSE: Mini Mental State Examination. BNT: Boston Naming Test. S-COWA: Semantic fluency. P-COWA: Phonologic fluency. ^1^
*p* < 0.05. ^2^
*p* < 0.01.

**Table 3 healthcare-11-02963-t003:** Multivariate linear regression analyses for factors associated with time reaction parameters.

Variable		B	β	t	95% CI		*p*-Value
MMSE (s)	Educational	1.177	0.278	3.195	0.448	1.906	0.002
	Dynamic Balance	−0.422	−0.203	−2.439	−0.765	−0.080	0.016
	Gait Speed	−1.353	−0.206	−2.409	−2.465	−0.242	0.017
BNT (score)	Educational	1.597	0.410	4.990	0.964	2.231	0.000
	Age	−0.119	−0.230	−3.023	−0.197	−0.041	0.003
	Gait Speed	−2.090	−0.346	−2.890	−3.521	−0.659	0.005
P-COWAT (words)	Educational	8.366	0.500	5.012	5.052	11.679	0.000
	Age	−0.512	−0.231	−2.676	−0.891	−0.132	0.009
	MNA-SF	−1.186	−0.167	−2.090	−2.313	−0.059	0.039
S-COWAT (words)	Educational	3.455	0.265	3.283	1.373	5.536	0.001
	Gait Speed	−5.957	−0.294	−2.201	−11.313	−0.601	0.030

B: standardized coefficient. β: standardized coefficient. CI: confidence interval. MMSE: Mini Mental State Examination. BNT: Boston Naming Test. P-COWAT: Phonologic fluency. S-COWAT: Semantic fluency.

## Data Availability

The data presented in this study are available on request from the corresponding author. The data are not publicly available due to the given the sensitive nature of the questions asked in this study and the necessary guarantees of privacy and confidentiality.

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
