# Peer review of "Influence of Nutritional Status and Physical Fitness on Cognitive Domains among Older Adults: A Cross-Sectional Study"

_healthcare, 2023, doi:10.3390/healthcare11222963_

Round 1

Reviewer 1 Report

Comments and Suggestions for Authors

In my opinion, the use of the MMSE is inappropriate for this target group as the test is designed to detect cognitive impairment. Furthermore, people with cognitive impairments were excluded. It is therefore not surprising that the MMSE score is in the normal range. Statistical comparisons with the MMSE are therefore questionable and not very meaningful.

 As described above, cognitive impairment is not present in the participants and is therefore not in line with the title of the article.

 Furthermore, nutritional status was assessed using only two questionnaires (MNA-SF and the EAT-10 dysphagia questionnaire). The first only gives information about the risk of malnutrition, the second is very subjective in its scoring. The MNA-SF is only a screening tool, whereas the MNA-LF, for example, would be more suitable for assessment. It is therefore questionable whether malnutrition can be adequately identified. In addition, biochemical parameters, BIA measurements, etc. could also be useful.

Table 1:

o The column headings in the header are inappropriate.

o The left column is not consistent and the educational status could be better designed (e.g. the 4 associated points could be indented at the bottom).

o The values in the columns refer to different things. For most columns, the right column is probably the standard deviation. However, when it comes to educational status, the values for men and women are 100%, just added up differently. This seems inconsistent and is not immediately clear.

Table 2 and associated text:

The statements in Table 2 are unclear. No such comparisons can or should be made within this small group of subjects.

 Overall, I would recommend that the part on cognition be removed from the article. A limitation to nutritional status and physical fitness with a more detailed description of the limitations would be desirable.

Author Response

Dr. Francisco Alvarez Salvago Department of Physiotherapy, Faculty of Health Sciences, European University of Valencia,

46112 Valencia, Spain

Tel +34-630734363

Editorial Reviewer 1

Healthcare

25th October 2023

Dear Reviewer 1,

First of all, we would like to thank you for giving us the possibility of addressing all the questions that arose during the review process and for your time dedicated to understanding and improving this scientific work. We think those comments have greatly improved the quality of this study. Thanks to your contributions and suggestions, together with all our responses, this scientific research is now much easier to read and understand. Please find below all the responses in a point-by-point fashion.

Please find below the answers to each of your contributions

 “1. In my opinion, the use of the MMSE is inappropriate for this target group as the test is designed to detect cognitive impairment. Furthermore, people with cognitive impairments were excluded. It is therefore not surprising that the MMSE score is in the normal range. Statistical comparisons with the MMSE are therefore questionable and not very meaningful.

 As described above, cognitive impairment is not present in the participants and is therefore not in line with the title of the article.

 Furthermore, nutritional status was assessed using only two questionnaires (MNA-SF and the EAT-10 dysphagia questionnaire). The first only gives information about the risk of malnutrition, the second is very subjective in its scoring. The MNA-SF is only a screening tool, whereas the MNA-LF, for example, would be more suitable for assessment. It is therefore questionable whether malnutrition can be adequately identified. In addition, biochemical parameters, BIA measurements, etc. could also be useful”.

Author response:

Thank you very much for your contributions. Taking into consideration your first comment, it is true that the MMSE is more appropriate for a target population consisting of individuals with dementia, but although it is less suited, it is also used for situations when there may be reasonable grounds for suspicion of some level of cognitive impairment, to assess the health status when it comes to any issues that might affect their cognitive function (Grigoletto F, Zappalà G, Anderson DW, Lebowitz BD. Norms for the Mini-Mental State Examination in a healthy population. Neurology. 1999 Jul 22;53(2):315-20. doi: 10.1212/wnl.53.2.315. PMID: 10430420.)  In the context of this study, we carried out the screening in order to determine whether any of the participants, since they are older adults, would exhibit some level of mild cognitive impairment, even if it was just small and not reported when asked, and to examine the implications of this factor.

Considering your contribution on the title of the article and fully agreeing with it, after a deep reflection we have determined that the most appropriate term to express the results obtained in the analysis of the BNT and COWAT, will be more appropriate to refer to the term Cognitive Domains in the enunciation of the manuscript (see Title).

In the context of nutritional status, we completely concur that the test employed might be substituted and supplemented with parameters obtained by blood analysis; for this reason, we have included it within the limitations of this study as per your proposal, line 15-17 and line 20 of the “Strengths and Limitations” section.

Given that the MNA-SF is a screening tool, the results should be interpreted with caution, and future investigations should consider using a more accurate assessment tool, such as The Mini Nutritional Assessment- Long Form (MNA-LF). For there to be sufficient evidence to further confirm the casual association founded, deeper research will be required to enlarge the sample size, follow-up with this cohort, add more information about food intake frequency and macronutrients, blood tests results, as well as validate these findings in other populations.”

 “2. Table 1:

  • The column headings in the header are inappropriate.
  • The left column is not consistent and the educational status could be better designed (e.g. the 4 associated points could be indented at the bottom).
  • The values in the columns refer to different things. For most columns, the right column is probably the standard deviation. However, when it comes to educational status, the values for men and women are 100%, just added up differently. This seems inconsistent and is not immediately clear.

Author response:

Thank you very much for your contributions once more. As mentioned in the comment related to the table 1. We have improved substantially the format and the appreciations in the table 1 – results section.

“3. Table 2 and associated text: The statements in Table 2 are unclear. No such comparisons can or should be made within this small group of subjects”.

Author response:

Thank you very much for your appreciation. Considering that in the case of the explanation in Table 2, these are correlation analyzes and we explain the correlations with a significance level of p<0.05. In addition, in the following analysis (table 3) we include these correlations in the multivariate linear regression to give even greater strength to the statistical analysis and comment on the associations between independent and dependent variables.

“4.  Overall, I would recommend that the part on cognition be removed from the article. A limitation to nutritional status and physical fitness with a more detailed description of the limitations would be desirable”.

Author response:

Thank you very much for this last comment.

We sincerely appreciate your comment. As previously declared the improvements on nutritional status have already been done as a result of your thoughtful and significant contributions on the first part of the present document, see line 15-17 and line 20 of the “Strengths and Limitations” section.

Given that the MNA-SF is a screening tool, the results should be interpreted with caution, and future investigations should consider using a more accurate assessment tool, such as The Mini Nutritional Assessment- Long Form (MNA-LF). For there to be sufficient evidence to further confirm the casual association founded, deeper research will be required to enlarge the sample size, follow-up with this cohort, add more information about food intake frequency and macronutrients, blood tests results, as well as validate these findings in other populations.”

 Also, in the limitations section following your guidance, this information was added (Strengths and Limitations line 10).

“The results should be interpreted with caution because no prior level of physical activity was taken into account for participation, and it's possible that those who engage in greater physical activity perform better on cognitive tests”, a reference of a study relating these two variables (and heart failure) has been also included to be more accurate (46).

The author responsible for correspondence is:

Dr. Francisco Álvarez Salvago Department of Physiotherapy, Faculty of Health Sciences, European University of Valencia, 46112 Valencia, Spain.

Tel +34-630734363; e-mail: [email protected]

Sincerely,

Francisco Álvarez Salvago

Reviewer 2 Report

Comments and Suggestions for Authors

Influence of nutritional status and physical fitness over cognitive impairment among older adults >60 years of age: a cross-sectional study.

The topic of the study is interesting since, although it is a work of associations, it can help guide future interventions to delay the onset of cognitive disease. Here are my comments:

Title: When the term older adults is used, it is understood that it refers to people over 60 years of age. I suggest removing ≥ 60 Years of Age as it is redundant and does not provide new information.

Keywords: It is recommended that keywords do not appear in the title. I suggest changing them. One possibility is to use Mesh terms.

2 Material and methods

Participants

Be careful with the terminology. keep older adults throughout the writing. Using senior citizen can lead to confusion.

Has the level of physical activity been considered? The relationship between physical condition and cognitive state is known, therefore whether a person is more or less active could be conditioning the results of the study.

Statistical analysis

Specify whether the Pearson or Spearman coefficient or both has been used.

3 Results

It would be interesting to include either the mean or the 95% CI. There is no information about the age range. That is, it is known that they are over 60 years old but nothing more and given that age is a determining factor in the variables analyzed, the results must be analyzed with caution since it is not the same that the sample is concentrated between 60 and 70 years old as We would talk about younger older adults than if they were distributed between 60 and 90 years of age, where there would be greater variability.

Explain why not all variables have been used in the multivariate analysis. That is, age and dynamic balance have been removed for some cognitive components. What was the reason? I think they should be included and provide more information about the relationships between variables.

4 Discussion and Conclusions

I think it can be expanded after analyzing the missing multivariate analyses. It would include some practical application, for example, that this analysis provides for the creation of interventions. Should we focus more on the nutritional aspect or the physical aspect? Do they have the same weight?

Author Response

Dr. Francisco Alvarez Salvago Department of Physiotherapy, Faculty of Health Sciences, European University of Valencia,

46112 Valencia, Spain

Tel +34-630734363

Editorial Reviewer 2

Healthcare

25th October 2023

Dear Reviewer 2,

First of all, we would like to thank you for giving us the possibility of addressing all the questions that arose during the review process and for your time dedicated to understanding and improving this scientific work. We think those comments have greatly improved the quality of this study. Thanks to your contributions and suggestions, together with all our responses, this scientific research is now much easier to read and understand. Please find below all the responses in a point-by-point fashion.

Please find below the answers to each of your contributions

 “1. Influence of nutritional status and physical fitness over cognitive impairment among older adults >60 years of age: a cross-sectional study”

  • The topic of the study is interesting since, although it is a work of associations, it can help guide future interventions to delay the onset of cognitive disease. Here are my comments:
  • Title: When the term older adults is used, it is understood that it refers to people over 60 years of age. I suggest removing ≥ 60 Years of Age as it is redundant and does not provide new information.
  • Keywords: It is recommended that keywords do not appear in the title. I suggest changing them. One possibility is to use Mesh terms.

Author response:

Thank you very much for your contributions, and for finding interesting the topic of the investigation.

In regard to the title, it has been modified, eliminating >60 years of age.

Additionally, keywords have been modified using Mesh terms.

 “2. Material and methods”

  • Participants: Be careful with the terminology. keep older adults throughout the writing. Using senior citizen can lead to confusion.
  • Has the level of physical activity been considered? The relationship between physical condition and cognitive state is known, therefore whether a person is more or less active could be conditioning the results of the study.
  • Statistical analysis: Specify whether the Pearson or Spearman coefficient or both has been used.

Author response:

Thank you very much for your contributions once more. As mentioned in your first recommendation, the term senior citizen has been eliminated to keep consistency throughout the text and to avoid possible confusion.

When it comes to your inquiry regarding the level of physical activity, it was not necessary for participants to engage in a particular level of physical activity in order to participate in the study because we wanted to use the tests to compare the relationships between the various levels of physical condition and their relationships with the other variables. Given your contribution, it will be noted in the limitations section that it was not assumed that participants had an initial degree of physical activity, therefore the findings should be interpreted with care. In the limitations section following your guidance, this information was added (Strengths and Limitations line 10).

“The results should be interpreted with caution because no prior level of physical activity was taken into account for participation, and it's possible that those who engage in greater physical activity perform better on cognitive tests”, a reference of a study relating these two variables (and heart failure) has been also included to be more accurate (46)”.

 In the statistical analysis the Pearson coefficient was used, and this information has been added in the Statistical Analyses section, line 10.

“3. Results”

  • It would be interesting to include either the mean or the 95% CI. There is no information about the age range. That is, it is known that they are over 60 years old but nothing more and given that age is a determining factor in the variables analysed, the results must be analysed with caution since it is not the same that the sample is concentrated between 60 and 70 years old as We would talk about younger older adults than if they were distributed between 60 and 90 years of age, where there would be greater variability.

  • Explain why not all variables have been used in the multivariate analysis. That is, age and dynamic balance have been removed for some cognitive components. What was the reason? I think they should be included and provide more information about the relationships between variables.

Author response:

Thank you very much for your appreciations. Considering that the part related to with the CI, the header of Table 3 includes this component. Regarding age, I concur with your statement and understand the variability. We have added information to the first line under "Results section" so that we can see the proximity in age range, which will help on the understanding of the sample.

“In a broader sense 100 senior citizens (mean 67.05 ± 5.17 years, between 60-75 years old) participated in the present study.”

In relation to age and dynamic balance, they were not eliminated from the analysis. In this analysis we included all the associations that were significant in the Pearson correlation and only these significant data are shown in Table 3 to demonstrate the associations with the greatest potential.

“4. Discussion and Conclusions!

  • I think it can be expanded after analysing the missing multivariate analyses. It would include some practical application, for example, that this analysis provides for the creation of interventions. Should we focus more on the nutritional aspect or the physical aspect? Do they have the same weight?

Author response:

            Thank you very much for your contribution. Considering your input, it has been decided to broaden the two sections, highlighting the glaring requirement for the scientific community to apply the findings to create future therapies that integrate dietary and physical health improvement to avoid cognitive decline. This information has been added on the “Discussion section” last three lines of the third paragraph:

“Also, the findings presented in this study could help the scientific community create intervention programs that focus on enhancing physical and nutritional health to prevent cognitive impairment at the clinical level.”

Moreover, a closing line for the “Conclusions section” was also added:  

“Additionally, at the clinical level, the research community may be able to develop intervention programs with the aid of the findings reported in this study.”

The author responsible for correspondence is:

Dr. Francisco Álvarez Salvago Department of Physiotherapy, Faculty of Health Sciences, European University of Valencia, 46112 Valencia, Spain.

Tel +34-630734363; e-mail: [email protected]

Sincerely,

Francisco Álvarez Salvago

Round 2

Reviewer 1 Report

Comments and Suggestions for Authors

Thank you for revising the manuscript and taking my comments into account.

Authors have explained all aspects in detail and reasons have been given.

The article has some weaknesses, but these are well presented in the limitations.

Reviewer 2 Report

Comments and Suggestions for Authors

Thank you for addressing all the comments.